# Prehospital Targeting of 1-Year Mortality in Acute Chest Pain by Cardiac Biomarkers

**DOI:** 10.3390/diagnostics13243681

**Published:** 2023-12-16

**Authors:** Daniel Zalama-Sánchez, Francisco Martín-Rodríguez, Raúl López-Izquierdo, Juan F. Delgado Benito, Irene Sánchez Soberón, Carlos del Pozo Vegas, Ancor Sanz-García

**Affiliations:** 1Servicio de Urgencias, Hospital Clínico Universitario de Valladolid, Gerencia Regional de Salud de Castilla y León (SACYL), 47007 Valladolid, Spain; dzalama@saludcastillayleon.es (D.Z.-S.); cpozove@saludcastillayleon.es (C.d.P.V.); 2Facultad de Medicina, Universidad de Valladolid, Gerencia de Emergencias Sanitarias, Gerencia Regional de Salud de Castilla y León (SACYL), 47007 Valladolid, Spain; 3Servicio de Urgencias, Hospital Universitario Rio Hortega de Valladolid, Gerencia Regional de Salud de Castilla y León (SACYL), 47007 Valladolid, Spain; rlopeziz@saludcastillayleon.es; 4Gerencia de Emergencias Sanitarias, Gerencia Regional de Salud de Castilla y León (SACYL), 47007 Valladolid, Spain; jdelgado@saludcastillayleon.es (J.F.D.B.); msanchezso@saludcastillayleon.es (I.S.S.); 5Grupo de Investigación en Innovación Tecnológica Aplicada a la Salud (Grupo ITAS), Facultad de Ciencias de la Salud, Universidad de Castilla la Mancha, 13071 Talavera de la Reina, Spain; ancor.sanz@gmail.com

**Keywords:** prehospital, acute chest pain, long-term mortality, troponin T, N-terminal pro B-type natriuretic peptide, D-dimer

## Abstract

The identification and appropriate management of patients at risk of suffering from acute chest pain (ACP) in prehospital care are not straightforward. This task could benefit, as occurs in emergency departments (EDs), from cardiac enzyme assessment. The aim of the present work was to derive and validate a scoring system based on troponin T (cTnT), N-terminal pro B-type natriuretic peptide (NT-proBNP), and D-dimer to predict 1-year mortality in patients with ACP. This was a prospective, multicenter, ambulance-based cohort study of adult patients with a prehospital ACP diagnosis who were evacuated by ambulance to the ED between October 2019 and July 2021. The primary outcome was 365-day cumulative mortality. A total of 496 patients fulfilled the inclusion criteria. The mortality rate was 12.1% (60 patients). The scores derived from cTnT, NT-proBNP, and D-dimer presented an AUC of 0.802 (95% CI: 0718-0.886) for 365-day mortality. This AUC was superior to that of each individual cardiac enzyme. Our study provides promising evidence for the predictive value of a risk score based on cTnT, NT-proBNP, and D-dimer for the prediction of 1-year mortality in patients with ACP. The implementation of this score has the potential to benefit emergency medical service care and facilitate the on-scene decision-making process.

## 1. Introduction

Acute chest pain (ACP) is a common cause of emergency medical service (EMS) activation, and identifying patients at risk of adverse outcomes for appropriate triage, support, and management is a challenge in the first steps [1]. Emergency departments (EDs) are provided with a different diagnostic toolbox to detect and discriminate patients with coronary chest pain. Cardiac enzymes, especially high-sensitivity cardiac troponin T (hs-cTnT), are routinely used and play a decisive role in identifying patients with a high probability of coronary ischemia and stratifying risk [2]. Nevertheless, in prehospital care, the identification and clinical stratification of patients with ACP are based on an objective and structured clinical assessment, focusing on the type, location, irradiation, intensity, and duration of pain and symptoms, together with a routine 12-lead electrocardiogram [3].

Novel point-of-care testing (POCT) has resulted in a significant change in out-of-hospital access to real-time diagnostic tests, providing a bedside mini-lab (allowing the assessment of venous gases, ions, renal profile, glucose, lactate, etc.). Despite not being a standard widespread procedure, the use of POCT in the normal workflow is rapidly beginning to be adopted in diverse EMSs [4] through the use of ultrasmall POCT devices that are rugged and user-friendly [5]. The next step has entailed the development of devices capable of providing cardiac enzyme analysis, analogous to the ED test, in a reduced delay. In this sense, biomarkers such as troponin T (cTnT), N-terminal pro B-type natriuretic peptide (NT-proBNP), or D-dimer are already available for implementation in prehospital care [6,7,8].

In addition, the ability of cTnT, particularly hs-cTnT, to diagnose and stratify patients with ACP is well known [9]. Similarly, the associations between elevated cTnT, NT-proBNP, or D-dimer levels and poor long-term outcomes in patients with ACP are notorious [10,11,12]. However, studies in prehospital care are limited and have almost always focused on short-term mortality [13,14]; moreover, there is a lack of precise knowledge of the role of these three biomarkers in relation to long-term mortality at the bedside.

The primary purpose of the present study was to derive and validate a scoring system encompassing prehospital cardiac biomarkers (cTnT, NT-proBNP, and D-dimer) to predict 1-year mortality in patients with ACP. The secondary purpose was to analyze the performance of the scoring system according to prehospital diagnosis of myocardial infarction code (MIC) vs. precordial pain with a normal electrocardiogram.

## 2. Materials and Methods

### 2.1. Study Design and Ethical Issues

A prospective, multicenter, ambulance-based cohort study was undertaken in patients with a prehospital ACP diagnosis who were evacuated by ambulance to the ED between November 2021 and October 2022 in the province of Valladolid (Spain), with a reference population of 524,204 inhabitants.

The study involved one case of advanced life support (ALS) (staffed by two emergency medical technicians, one emergency registered nurse and one physician), fourteen basic life supports (BLS) (staffed by two emergency medical technicians) and two tertiary university hospitals. Out-of-hospital and in-hospital facilities were operated and controlled by the public health system (SACYL). Advanced life support is based on internationally accepted clinical practice guidelines [3].

This observational study was conducted after approval by the respective local ethics committees (PI217-20 and PI-GR-20-1970) and registered in the World Health Organization’s International Clinical Trials Registry Platform (ICTRP), available online (https://doi.org/10.1186/ISRCTN48326533 accessed on 28 November 2023), and we followed the strengthening the reporting of observational studies in epidemiology (STROBE) [15] statement. All participants read and signed the informed consent form.

### 2.2. Participants

Adults (>18 years) with unselected prehospital ACP diagnoses who were subsequently referred to the ED were included in the study on a 24/7 basis.

Participants were recruited among emergency calls to the 1-1-2 call center. An emergency manager geopositioned the call and gathered the affiliation data. Next, after a brief guided survey, a medical or nurse manager dispatched an ALS to the site. Finally, if the prehospital diagnosis was ACP, the participant was screened for eligibility. The inclusion of cases can be from a reverse pathway. Sometimes, the coordinating center dispatches a BLS to the scene, and on-site emergency medical technicians request evaluation by the ALS, as an alternative way of recruiting participants. All patients were always screened by an ALS physician and subsequently referred to the ED (either ALS or BLS derivation).

Minors, pregnant women (evident or probable), patients with unobserved cardiac arrest, traumatic ACP, refusal of hospital referral, failure to collect the three prehospital biomarkers, no informed consent (from the patient or a relative or legal guardian), and patients unable to fulfill the 365-day follow-up period were excluded.

### 2.3. Outcomes

The primary outcome was 365-day cumulative mortality (including in-hospital, out-of-hospital, and all-cause mortality). The secondary objective of this work was related to the prehospital diagnosis (presenting MIC or precordial pain with normal electrocardiogram), which cannot be considered an outcome but is presented here as part of the objectives.

The electrocardiographic criteria for an ACP to trigger MIC include (i) ST-segment elevation, measured at the J-point, in two contiguous leads ≥ 0.1 mv (≥0.2 mv in leads V2-V3); (ii) left or right bundle branch block, either new-onset or previously reported; and (iii) ventricular paced rhythm.

### 2.4. Predictors and Data Collection

The epidemiological variables (sex at birth, age, living area (urban or rural), and type of ambulance), on-scene vital signs (respiratory rate, oxygen saturation, systolic and diastolic blood pressure, heart rate, temperature, and Glasgow coma scale score), and prospective on-scene biomarkers (cTnT, NT-proBNP, and D-dimer) were obtained upon first contact with the patient by the ALS emergency registered nurse. Electrocardiography and venous line access were also performed by the ALS nurse. The ALS physician documented the rhythm, ST-segment alterations, and special advanced life support steps performed on-scene or en route, e.g., advanced airway management, defibrillation, cardioversion, pacing, and vasoactive agents. Finally, the physician tagged the case as an ACP or MIC. Oxygen saturation, blood pressure, heart rate, and temperature were measured with a LifePAK^®^ 15 monitor-defibrillator (Physio-Control, Inc., Redmond, WA, USA). Cardiac biomarkers were assessed with CARDIAC proBNP+ immunoassay test strips using the cobas h232 POC instrument (Roche Diagnostics, Mannheim, Germany) following the manufacturer’s instructions [16].

After a 1-year follow-up period, an associate investigator, by review of the electronic medical records, collected the main outcome (365-day mortality) and the 17 comorbidities needed to calculate the age-adjusted Charlson comorbidity index (ACCI). Finally, intensive care unit admission, fibrinolysis, percutaneous coronary intervention, emergent cardiovascular surgery, mechanical ventilation, and/or vasoactive agent use data were collected at the hospital level.

### 2.5. Data Analysis

Medians and interquartile ranges (25th–75th percentiles) were used for quantitative variables due to nonnormal distributions (Shapiro–Wilk and Kolmogorov–Smirnov tests), and the comparison of means was assessed by the Mann–Whitney U test. Categorical variables are described by their absolute frequencies and 95% confidence intervals (95% CIs), and for 2 × 2 contingency tables or/and contrasts of proportions to determine the associations or dependency relationships of categorical variables, the chi-square test (or, if necessary, Fisher’s exact test) was used.

The score was obtained from the combination of D-dimer, NT-proBNP, and cTnT. Patients were randomly split into derivation (two-thirds of the total sample) and validation (one-third of the total sample) cohorts. The variables were subsequently categorized based on their relationship with the outcome variable and the previously described ranges [17,18]. The weight of each predictor range was derived from the β-coefficients (rounded to integers) of a logistic regression model. The final score was calculated as the sum of each patient’s score for each variable. The performance of the score, each score component, and each subset of patients according to confounding variables was assessed by using the area under the curve (AUC) of the receiver operating characteristic (ROC) curve and the calibration curve. Further details on the score development and validation methods can be found in the Appendix A.

Further details on the data collection, missing values, and sample size calculations can be found in the Appendix A. All the statistical analyses were performed using our own codes and base functions in R, version 4.0.3 (http://www.R-project.org, accessed on 1 October 2023

## 3. Results

A total of 496 patients were included in the final cohort (Figure 1). The median age was 71 years (IQR: 51–80), and 38.3% (190 patients) were females. Approximately 24.2% (120 patients) were from rural areas, and 71.8% (365 patients) were transferred to ALS. The cumulative mortality rates up to 365 days were 12.1% (60 patients), 51.6% (31 patients) and 48.3% (29 patients) of the total in-hospital mortality.

The median cTnT, NT-proBNP, and D-dimer levels were significantly greater in nonsurvivors than in survivors (*p* < 0.001 for all patients) (Table 1). Among nonsurvivors, the incidence of tachyarrhythmias, ST-segment elevation, and associated on-scene and hospital advanced life support procedures were greater. The nonsurvivors had an ACCI of 3 points (IQR: 2–5), with a hospital admission rate of 86.7% (52 patients) and ICU admissions of 61.7% (36 patients), whereas survivors had a comorbidity burden of 2 points (IQR: 1–3), an admission rate of 57.1% (249 patients), and ICU admissions of 36% (157 patients) (Table 1).

Among the diagnostic agents, 71% (352 patients) presented with ACP without an MIC, and 29% (144 patients) presented with an MIC. The associated mortalities were 9.9% (35 patients) and 17.9% (25 patients) for ACP without MIC and for ACP with MIC, respectively. An estimated 70.1% (101 patients) of the patients with MICs showed ST-segment elevation, with cTnT (180 ng/L) and NT-proBNP (432 pg/mL) levels; these data are significantly superior to the results obtained with PCI (0 ng/L and 201 pg/mL, respectively). Patients with MICs had higher rates of advanced life support maneuvers, both out-of-hospital and in-hospital, with admission rates of 95.8% (138 patients), ICU admission rates of 86.1% (124 patients), and percutaneous coronary intervention rates of 81.3% (117 patients) (Table 2).

The resulting score that combined all three parameters can be found in Table 3. A lower mortality risk occurred for those with scores between 0 and 1 point, and the highest mortality risk occurred for those with scores between 3 and 5 points, particularly in the 3–4 points group (Figure 2a). The AUC reached 0.802 (95% CI: 0718–0.886) (Figure 2b).

When considering different confounding factors, the performance of the score was as follows: for patients with MICs, the area under the curve (AUC) was 0.875 (95% CI: 0778–0.973) (Figure 3a); for those with ACPs, the AUC was 0.745 (95% CI: 0.617–0.872) (Figure 3b). For males, the 95% CI was 0.823 (0.722–0.924) (Figure 3c), and for females, the 95% CI was 0.792 (0.675–0.91) (Figure 3d). Regarding the age range, the area under the curve (AUC) was 0.737 (95% CI = 0.385–1) (Figure 3e), 0.89 (95% CI = 0.755–1) (Figure 3f), and 0.687 (95% CI = 0.541–0.833) (Figure 3g) for 18 to 49 years, 50 to 74 years, and >74 years, respectively. All these results were supported by the calibration curves, which can be found in Appendix A.

To determine the predictive power of each score component alone, the same procedure was used for cTnT, NT-proBNP and D-dimer, for which the AUCs were 0.696 (95% CI = 0.609–0.782) (Figure 4a), 0.746 (95% CI = 0.669–0.859) (Figure 4b), and 0.538 (95% CI = 0.432–0.644) (Figure 4c), respectively. All these results were supported by the calibration curves, which can be found in Appendix A.

## 4. Discussion

To our knowledge, this is the first ambulance-based cohort study to derive a risk score to predict 365-day mortality in adults with ACP using prehospital cardiac biomarkers. The score ranges from 0 to 5, with a higher 1-year mortality risk as the score increases.

Current practice in emergency medicine focuses on the combination of different biomarkers to enhance the predictive capacity of biomarkers. Compared to previous hospital-based studies [19,20,21], our study provides a novel approach by developing a scoring system that combines cTnT, NT-proBNP, and D-dimer levels measured in the prehospital setting to predict 1-year mortality in patients with ACP. The novel score was based on the combination of three biomarkers supported by previous scientific evidence: cTnT, NT-proBNP, and D-dimer. Individually, they are well known for evaluating patients with CVD, demonstrating the association between the elevation of these biomarkers and 1-year mortality.

Elevated NT-proBNP levels imply a worse prognosis across the spectrum of acute coronary syndromes [19]. In addition, they have been shown to be a predictor of long-term mortality in patients with coronary syndrome [22] and to be an independent predictor of very long-term all-cause mortality [20]. Similarly, elevated troponin levels are associated with increased mortality, independent of age [10]. Troponin T can predict patients with high all-cause mortality at 1 and 5 years and is also associated with a risk of rehospitalization related to acute heart failure, adding prognostic value to NT-proBNP [21,23]. Finally, an association between increased D-dimer concentration and an increase in major adverse cardiovascular events, such as heart failure, malignant arrhythmias, and death, has been reported in hospitals. Elevated D-dimer values predict the long-term risk of arterial and venous events, cardiovascular disease mortality, and mortality unrelated to cardiovascular disease and cancer, independent of other risk factors. A consistently elevated D-dimer level was found to be an independent predictor of long-term all-cause mortality [24].

Among those who were single-handed, the best predictive ability for 1-year mortality was exhibited by NT-proBNP (AUC = 0.764), followed by cTnT (AUC = 0.696) and D-dimer (AUC = 0.583). The combination of these three biomarkers achieves synergistic effects that improve the ability of the scoring systems to perform risk prediction and stratify patients with ACP into high- and low-risk categories [25]. The results obtained reasonably good performance, with an AUC of 0.802. In addition, a stratified analysis according to clinical variables, such as major adverse cardiac events, sex, and age, was carried out, providing additional information on the clinical usefulness of the scoring system in various subcohorts. However, in the subanalysis by age group, disparities were found in the predictive capacity, differences that can be explained by the changes generated by chronological age, by the comorbidities and by the level of frailty, making it very difficult to characterize older adults.

Clinical implementation of this score has the potential to improve decision-making in prehospital care. Currently, it is not easy to determine the 1-year outcomes of patients since hidden pathologies or secondary consequences can emerge after an acute event. Therefore, extending the clinical decision-making process for early warning scores from short-term to 1-year outcomes could be particularly helpful for planning long-term care once the acute phase of the disease has ended or even activating follow-up programs in the health system since the patient is potentially fragile. As a result, faster referral to specialized units, the administration of specific therapies or the consideration of appropriate targeted interventional interventions may be facilitated. The identification and clinical stratification of patients with chest pain in prehospital care remain major challenges [26]. Therefore, the inclusion of prehospital cardiac biomarkers obtained at the bedside (during the hyperacute phase) provides a valuable opportunity for timely recognition of high-risk patients, broadening diagnostic capabilities and improving clinical decision-making [27,28,29,30].

This study has several limitations. First, the study was conducted in a specific clinical context and included a limited sample of patients. Therefore, further validation in different populations and settings is needed to confirm the generalized utility of this scoring system. Furthermore, although relevant biomarkers were included, other clinical and laboratory factors that could contribute to the prediction of 1-year prognosis in patients with ACP were not considered in this study. Similarly, only patients treated for ALS were included. However, it is important to note that the applicability of this score in other settings, such as BLS or health centers, requires further validation studies to assess its efficacy in that particular setting. Second, the data extractors were not blinded. To avoid cross-contamination, the EMS providers were not aware of the joint utility of the three biomarkers, nor did they have any access to the follow-up data. Similarly, the hospital research associates were unaware of the prehospital care assessment and care data. Only the principal investigator and the data manager had full access to the master database. Third, our study explored the predictive capacity of the single-score components but not whether the combination of all three biomarkers was better than the combination of any of two. In fact, D-dimer presented an AUC close to 0.5, suggesting that D-dimer has limited predictive power on its own. However, it should be noted that the combination of the three score components results from logistic regression, which considers the variables globally, allowing us to study their association. Therefore, as can be observed in the results, the predictive power of logistic regression always benefits from including all the variables in the model. Finally, the study was conducted before the COVID-19 pandemic and during the progression of the pandemic. During the peak phase of the outbreak, the demand for care was known to decline dramatically for reasons other than those strictly due to the pandemic. Evidence shows that activations due to time-dependent processes suffer a spectacular drop in incidence, not necessarily indicating that the number of cases has decreased but rather that the population did not want to go to hospitals out of fear [31,32]. As a consequence, by the end of 2020, an overall mortality rate was experienced that was not solely attributable to COVID-19. Consequently, it is necessary to interpret the incidence data with caution.

## 5. Conclusions

In summary, our study provides promising evidence for the value of three prehospital cardiac biomarkers (cTnT, NT-proBNP, and D-dimer) for the prediction of 1-year mortality in patients with ACP. This could lead to important advances in risk stratification, and 1-year mortality estimation in patients with ACP has been performed with the development of a scoring system based on prehospital cardiac biomarkers. The resulting score could be useful in the management of these patients in prehospital care, allowing for better risk stratification and more appropriate allocation of medical resources. This approach has the potential to benefit EMS care and facilitate the on-scene decision-making process.

## Figures and Tables

**Figure 1 diagnostics-13-03681-f001:**
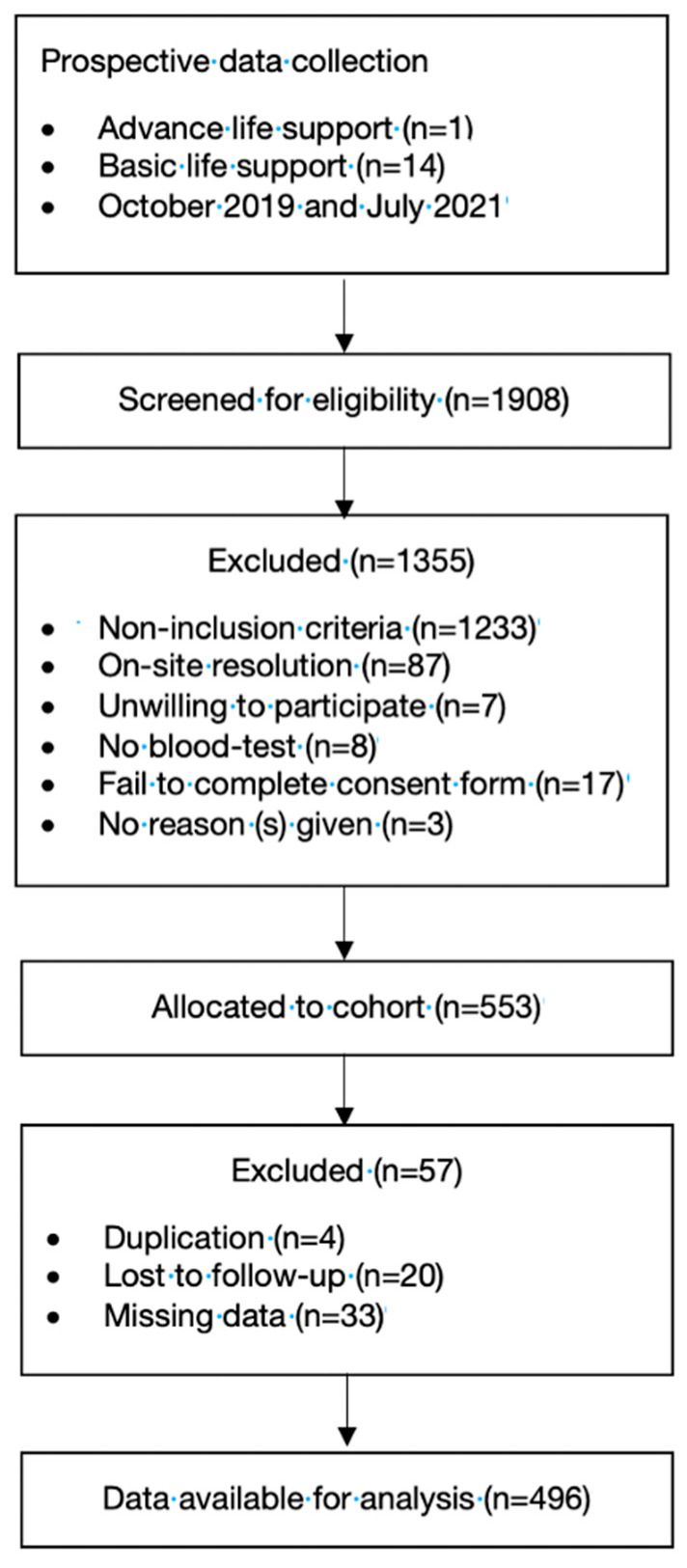
Flowchart of the study population.

**Figure 2 diagnostics-13-03681-f002:**
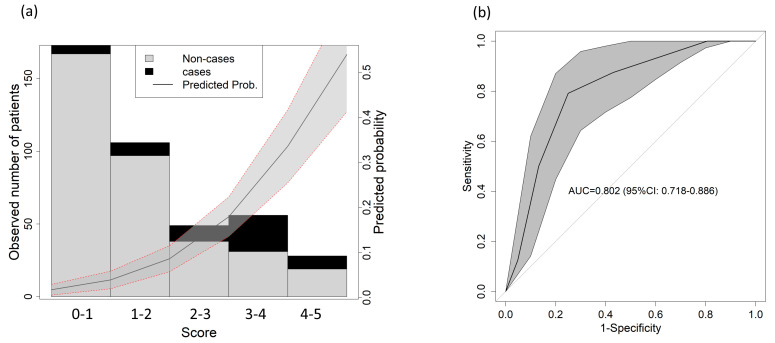
Score predictive validity for 1-year mortality. (**a**) Observed distribution of the outcome based on the score. The solid line shows the predicted probability of the outcome; the gray shadowed area shows the 95% confidence interval. (**b**) AUC for 1-year mortality in the validation cohort.

**Figure 3 diagnostics-13-03681-f003:**
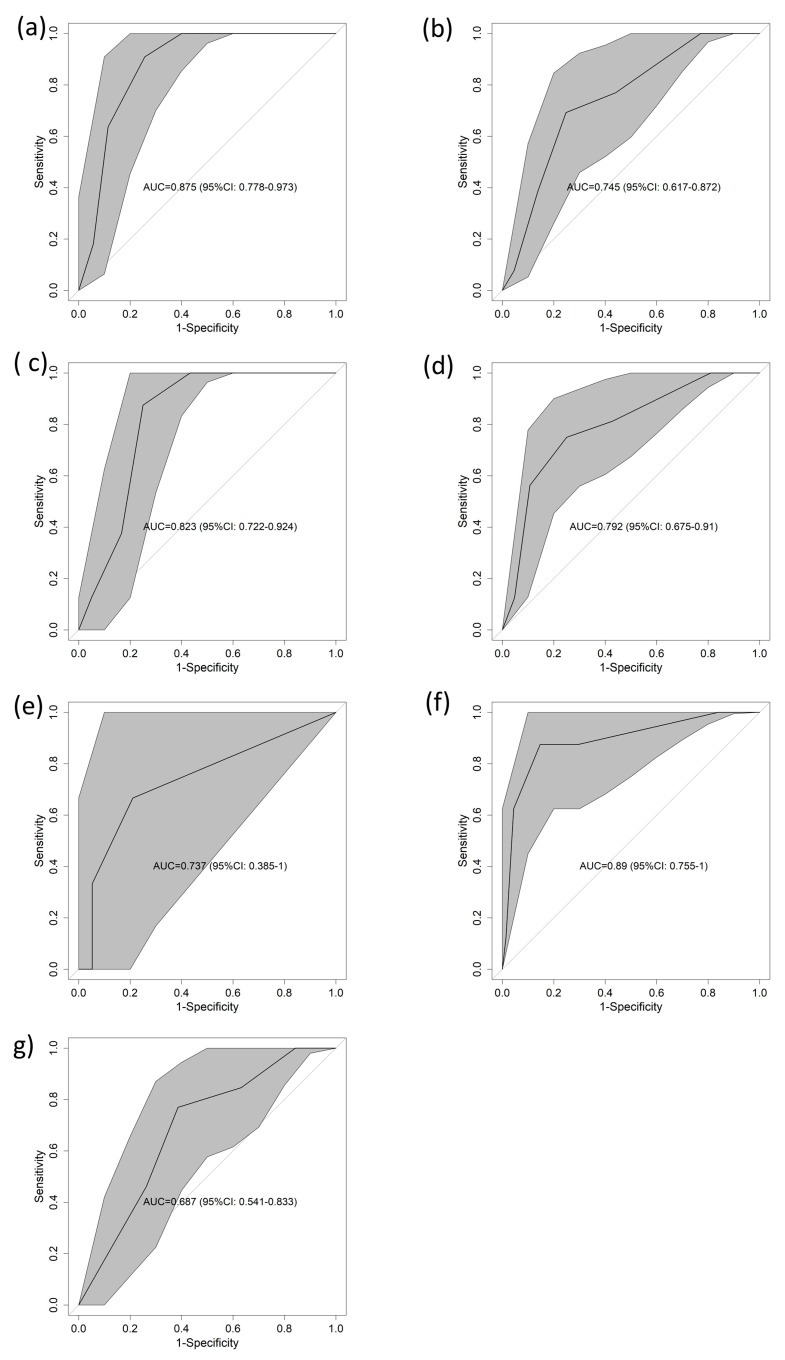
AUCs for 1-year mortality in the validation cohort according to different subsets. (**a**) ACP, (**b**) MIC, (**c**) female, (**d**) male, (**e**) 18 to 49 years, (**f**) 50 to 74 years, (**g**) over 74 years. Abbreviations: AUC—area under the curve of the receiver operating characteristic; CI—confidence interval.

**Figure 4 diagnostics-13-03681-f004:**
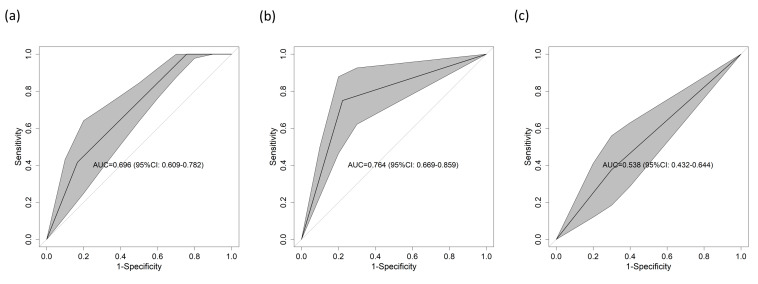
AUC for 1-year mortality in the validation cohort according to the different parameters included in the score considered individually. (**a**) cTnT, (**b**) NT-proBNP, and (**c**) D-dimer. All the tables had *p* values < 0.05, except for (**c**). Abbreviations: AUC—area under the curve of the receiver operating characteristic; CI—confidence interval.

**Table 1 diagnostics-13-03681-t001:** Baseline patient characteristics, principal outcomes, and other determinants of 365-day mortality.

Variables	Total	Survivors	Nonsurvivors	*p* Value ^2^
No. (%) with data ^1^	496	436 (87.9)	60 (12.1)	NA
Epidemiological variables				
Sex at birth, female	190 (38.3)	172 (39.4)	18 (30)	0.159
Age, year	71 (51–80)	71 (57–80)	75 (65–85)	0.004
Age groups, year				
18–49	63 (12.7)	58 (13.3)	5 (8.3)	0.061
50–74	231 (46.6)	207 (47.5)	24 (40)	
>75	202 (40.7)	171 (39.2)	31 (51.7)	
Zone, rural	120 (24.2)	106 (24.3)	14 (23.3)	0.869
Transfer, ALS	365 (71.8)	305 (70)	51 (85)	0.015
On-scene vital signs				
RR, breaths/min	16 (14–19)	16 (14–19)	18 (14–27)	<0.001
SpO2, %	97 (95–98)	97 (95–98)	94 (85–97)	<0.001
SBP, mmHg	144 (122–168)	146 (125–169)	129 (89–156)	<0.001
DBP, mmHg	82 (69–95)	83 (77–96)	65 (46–81)	<0.001
Heart rate, beats/min	78 (65–95)	77 (65–93)	86 (68–124)	0.004
Temperature, °C	36 (35.8–36.8)	36.1 (35.8–36.5)	36 (35.6–36.7)	0.243
Glasgow coma scale, points	15 (15–15)	15 (15–15)	15 (7–15)	0.001
Baseline cardiac rhythm				
Sinus	291 (51.7)	275 (63.1)	16 (26.7)	<0.001
Tachycardia ^3^	149 (30)	112 (25.7)	37 (67.1)	
Bradycardia ^4^	46 (9.3)	41 (9.8)	5 (8.3)	
Pacemaker	10 (2)	8 (1.8)	2 (3.3)	
ST elevation ^5^	113 (22.8)	95 (21.8)	18 (30)	0.594
Prehospital diagnosis				0.021
Acute chest pain	352 (71)	317 (72.7)	35 (58.3)	
Myocardial infarction code	144 (29)	119 (27.3)	25 (41.7)	
Cardiac biomarkers				
Troponin T, ng/L	47 (0–109)	42 (0–91)	112 (57–336)	<0.001
NT-proBNP, pg/mL	235 (63–997)	221 (0–643)	2131 (932–5583)	<0.001
D-dimer, microg/mL	410 (220–753)	352 (200–680)	954 (414–2260)	<0.001
Support on-scene				
Mechanical ventilation	23 (4.6)	4 (0.9)	19 (31.7)	<0.001
Defibrillation	14 (2.8)	3 (0.7)	11 (18.3)	<0.001
Cardioversion	11 (2.2)	6 (1.4)	5 (8.3)	0.001
Transcutaneous pacemaker	9 (1.8)	7 (1.6)	2 (3.3)	0.348
Vasoactive agents	25 (5)	6 (1.4)	19 (31.7)	<0.001
Hospital outcome				
ACCI, points	2 (1–4)	2 (1–3)	3 (2–5)	0.004
Inpatient	301 (60.7)	249 (57.1)	52 (86.7)	0.001
Hospitalization time, days	3 (0–6)	0 (0–1)	2 (0–14)	0.069
Fibrinolysis	17 (3.4)	8 (1.8)	9 (15)	<0.001
PCI	193 (38.1)	163 (37.1)	30 (50)	0.061
ECS	9 (1.8)	8 (1.8)	1 (1.7)	0.927
Mechanical ventilation	41 (8.3)	17 (3.9)	24 (40)	<0.001
Vasoactive agents	49 (9.9)	20 (4.6)	29 (48.3)	<0.001
ICU-admission	194 (39.1)	157 (36)	37 (61.7)	<0.001

Abbreviations: NA—not applicable; RR—respiratory rate; SpO2—oxygen saturation; SBP—systolic blood pressure; DBP—diastolic blood pressure; NT-proBNP—N-terminal pro B-type natriuretic peptide; ACCI—age-adjusted Charlson comorbidity index; PCI—percutaneous coronary intervention; ECS—emergent cardiovascular surgery; ICU—intensive care unit. ^1^ Values are expressed as the total number (percentage) and median (25th–75th percentile), as appropriate. ^2^ The Mann–Whitney U test or chi-squared test was used as appropriate. ^3^ Tachycardia rhythm included sinus tachycardia (87, 18.5%), atrial fibrillation (49, 9.9%), atrial flutter (1, 0.2%), supraventricular tachycardia (4, 0.8%), and ventricular tachycardia (8, 1.6%). ^4^ The bradycardia rhythm included sinus bradycardia (29, 5.8%), first-degree atrioventricular (AV) block (12, 2.4%), Mobitz type I 2nd-degree AV block (1, 0.2%), Mobitz type II 2nd-degree AV block (2, 0.4%), and third-degree AV block (2, 0.4%). ^5^ Patients had a normal ST (228, 46%), ST segment depression (53, 10.7%), a peaked T wave (23, 4.6%), a negative T wave (54, 10.9%), or a Q wave (25, 5%).

**Table 2 diagnostics-13-03681-t002:** Baseline patient characteristics, principal outcomes, and other determinants according to myocardial infarction code vs. acute chest pain.

Variables	Acute Chest Pain	Myocardial Infarction Code	*p* Value ^2^
No. (%) with data ^1^	352 (71)	144 (29)	NA
Epidemiological variables			
Sex at birth, female	149 (42.3)	41 (28.5)	0.004
Age, year	73 (59–83)	67 (56–77)	0.119
Zone, rural	77 (21.8)	43 (29.9)	0.061
Transfer, ALS	137 (38.9)	141 (97.9)	<0.001
On-scene vital signs			
RR, breaths/min	16 (14–20)	16 (13–18)	0.057
SpO2, %	97 (95–98)	96 (94–98)	0.001
SBP, mmHg	145 (125–170)	140 (115–165)	0.006
DBP, mmHg	82 (70–95)	80 (67–99)	0.248
Heart rate, beats/min	78 (67–95)	77 (60–96)	0.803
Temperature, °C	36.1 (35.8–36.5)	36 (35.8–36.4)	0.132
Glasgow coma scale, points	15 (15–15)	15 (15–15)	0.186
Baseline cardiac rhythm			
Sinus	206 (58.5)	85 (59)	0.758
Tachycardiac ^3^	111 (31.5)	38 (26.4)	
Bradycardiad ^4^	26 (7.4)	20 (13.9)	
Pacemaker	9 (2.6)	1 (0.7)	
ST elevation	12 (3.4)	101 (70.1)	<0.001
Cardiac biomarkers			
Troponin T, ng/L	0 (0–58)	180 (57–431)	<0.001
NT-proBNP, pg/mL	201 (0–702)	432 (199–1612)	0.001
D-dimer, microg/mL	331 (199–751)	469 (293–753)	0.069
Support on-scene			
Mechanical ventilation	4 (1.1)	19 (13.2)	<0.001
Defibrillation	1 (0.3)	13 (9)	<0.001
Cardioversion	7 (2)	4 (2.8)	0.589
Transcutaneous pacemaker	3 (0.9)	6 (4.2)	0.012
Vasoactive agents	7 (2)	18 (12.5)	<0.001
Hospital outcome			
ACCI, points	2 (1–4)	2 (1–3)	0.021
Inpatient	163 (46.3)	138 (95.8)	<0.001
Hospitalization time, days	1 (0–6)	5 (3–8)	0.011
Fibrinolysis	8 (2.3)	9 (6.3)	0.027
PCI	76 (21.6)	117 (81.3)	<0.001
ECS	5 (1.4)	4 (2.8)	0.305
Mechanical ventilation	15 (4.3)	26 (18.1)	<0.001
Vasoactive agents	18 (5.1)	31 (21.5)	<0.001
ICU-admission	70 (19.9)	124 (86.1)	<0.001

Abbreviations: NA—not applicable; ALS—advanced life support; RR—respiratory rate; SpO2—oxygen saturation; SBP—systolic blood pressure; DBP—diastolic blood pressure; NT-proBNP—N-terminal pro B-type natriuretic peptide; ACCI—age-adjusted Charlson comorbidity index; PCI—percutaneous coronary intervention; ECS—emergent cardiovascular surgery; ICU—intensive care unit. ^1^ Values are expressed as the total number (percentage) and median (25th–75th percentile), as appropriate. ^2^ The Mann–Whitney U test or chi-squared test was used as appropriate. ^3^ Tachycardia rhythm includes sinus tachycardia, atrial fibrillation, atrial flutter, supraventricular tachycardia, and ventricular tachycardia. ^4^ The bradycardia rhythm included sinus bradycardia, first-degree atrioventricular (AV) block, Mobitz type I 2nd-degree AV block, Mobitz type II 2nd-degree AV block, and third-degree AV block.

**Table 3 diagnostics-13-03681-t003:** Prehospital cardiac biomarker risk score.

Variables	0	1	2
Troponin T, ng/L	≤39	40–139	≥140
NT-proBNP, pg/mL	≤999		≥1000
D-dimer, microg/mL	≤199	200–999	≥1000

Abbreviations: NT-proBNP—N-terminal pro B-type natriuretic peptide.

## Data Availability

The data from the study are available to other researchers upon reasonable request to the corresponding author.

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
