# Peer review of "Prehospital Targeting of 1-Year Mortality in Acute Chest Pain by Cardiac Biomarkers"

_diagnostics, 2023, doi:10.3390/diagnostics13243681_

Round 1
Reviewer 1 Report (Previous Reviewer 1)
Comments and Suggestions for Authors
This is a repeat peer review of the manuscript. The authors has made considerable changes in accordance to their rebuttal. I have the following comments:
Major:
1) The authors state that the cohort was split between derivation and validation cohorts using "the R package “caret” was used, since it allows to perform the split by maintaining the proportion of outcomes". I am fine with the methodology. However, in no place in the methods were this details expounded upon. For reproducibility and accountabilty, please report all detailed methods accordingly in-text.
2) Calibration curves - Good that it has been moved to supplementary data, it looks cleaner now. For easier understanding, all acronyms/abbreviations in these graphs should also be spelt out for the audience (even in data supplements) in the figure legend as pointed out in the rebuttal.
3) Limitations - while I understand that this study was not set out to compare 2 out of 3 of the biomarkers as the authors have explained, however, this should still be discussed as a limitation here in-text. Fig 4C displayed an AUC of 0.538, non-sig. Why may this be the case? Is D-dimer really predictive at all in your cohort studies or was there differences in cohorts from other studies that may explain if D-dimer may be less powerful here? However, this was not discussed at all and I have to repeat that potentially important details were "skimmed over".
Minor:
1) Abstract line 22-23 - there's now a duplication of "Prospective". Additionally, page 2 line 70, the duplication of "multicenter" and "prospective" is still in the same sentence.
2) IRB approval - If one IRB approval was functionally applied across both institutional sites and no secondary IRB was required, this should be explained in the IRB statement. Some countries will require IRB approvals specifying the exact multi-centers to be validated by the IRB of secondary institutions carrying out the work and can have very different regulations between institutions and inter-nationally. It is good to be specific for accountability.
3) Figure 1 seems like there was some formatting issue with tracked changes. Please remove the old flowchart.
4) Suggest to give the detailed method for score development and validation a subheading "Supplemental Methods" in the data supplement, and reference to this in-text on page 4 line 149 for clarity.
Comments on the Quality of English LanguageMinor grammatical proofreading will be required to improve sentence structure.
Author Response
Dear Editor,
We would like to acknowledge the thorough revision of our work by the editor and reviewers. We also thank you for the opportunity to resubmit our work. We have responded to all the reviewers’ comments. Below, please find point-by-point responses coloured in red. All the comments that needed a change in the manuscript have been included in the main document and are also colored in red. Additionally, the whole text has been reviewed by the AJE editing service.
We believe that the manuscript has been substantially improved, and we hope that it can be considered for publication.
Kind regards,
Francisco Martín-Rodríguez on behalf of all the authors
Reviewer #1
This is a repeat peer review of the manuscript. The authors has made considerable changes in accordance to their rebuttal. I have the following comments:
Major:
1) The authors state that the cohort was split between derivation and validation cohorts using "the R package “caret” was used, since it allows to perform the split by maintaining the proportion of outcomes". I am fine with the methodology. However, in no place in the methods were this details expounded upon. For reproducibility and accountabilty, please report all detailed methods accordingly in-text.
Response: We agree with the reviewer that this point should be clarified in the text. This information has been included in the supplementary information.
2) Calibration curves - Good that it has been moved to supplementary data, it looks cleaner now. For easier understanding, all acronyms/abbreviations in these graphs should also be spelt out for the audience (even in data supplements) in the figure legend as pointed out in the rebuttal.
Response: We agree with the reviewer that this point should be clarified in the text. The acronyms/abbreviations of the graphs have been included in the figure captions.
3) Limitations - while I understand that this study was not set out to compare 2 out of 3 of the biomarkers as the authors have explained, however, this should still be discussed as a limitation here in-text. Fig 4C displayed an AUC of 0.538, non-sig. Why may this be the case? Is D-dimer really predictive at all in your cohort studies or was there differences in cohorts from other studies that may explain if D-dimer may be less powerful here? However, this was not discussed at all and I have to repeat that potentially important details were "skimmed over".
Response: We have added that point to the limitations section at the end of the discussion. We have also included our response to this question about D-dimer levels.
“Third, our study explored the predictive capacity of the single score components but not whether the combination of all three biomarkers was better than the combination of any of two. In fact, D-dimer presented an AUC close to 0.5, suggesting that D-dimer has limited predictive power on its own. However, it should be noted that the combination of the three score components results from logistic regression, which considers the variables globally, allowing us to study their association. Therefore, as can be observed in the results, the predictive power of logistic regression always benefits from including all the variables in the model.”
Minor:
1) Abstract line 22-23 - there's now a duplication of "Prospective". Additionally, page 2 line 70, the duplication of "multicenter" and "prospective" is still in the same sentence.
Response: Solved
2) IRB approval - If one IRB approval was functionally applied across both institutional sites and no secondary IRB was required, this should be explained in the IRB statement. Some countries will require IRB approvals specifying the exact multi-centers to be validated by the IRB of secondary institutions carrying out the work and can have very different regulations between institutions and inter-nationally. It is good to be specific for accountability.
Response: Despite not being necessary in our country to provide both IRB codes, we agree with the reviewer; therefore, the IRB codes of the two hospitals have been included.
3) Figure 1 seems like there was some formatting issue with tracked changes. Please remove the old flowchart.
Response: Solved
4) Suggest to give the detailed method for score development and validation a subheading "Supplemental Methods" in the data supplement, and reference to this in-text on page 4 line 149 for clarity.
Response: Done

Reviewer 2 Report (New Reviewer)
Comments and Suggestions for Authors
I reviewed with interest the manuscript by Daniel Zalama-Sánchez et al, “Prehospital targeting of long-term mortality in acute chest pain by cardiac biomarkers.” In this article, the authors showed that a risk score based on prehospital cTnT, NT-proBNP, and D-dimer was able to predict long-term mortality in patients with acute chest pain.
This is probably an interesting result that expands the possibilities of prehospital diagnostics. However, during the review, I had questions and comments to which I would like to receive answers from the authors:
1. First of all, I have doubts about the basic concept of the work. In patients with acute chest pain, rapid diagnosis is required, and if acute coronary syndrome is detected, emergency treatment is required. Therefore, prehospital diagnostics should be aimed precisely at this, for example, this was implemented in the AROMI study (ref. 1-32, see below). Identifying prehospital predictors of 1-year mortality in patients with acute chest pain seems far-fetched. If there is no acute coronary pathology, then this can be safely done during an outpatient examination in primary care. I also disagree with the use of the term “long-term mortality” in relation to annual mortality.
2. The authors submitted the text of the manuscript with highlighted changes after editing (apparently, according to previous reviews). This is unnecessary at this stage.
3. The purpose of duplicating the block diagram (Fig. 1) is unclear to me.
4. In Figure 2a, visually the highest percentage of mortality was in the group with 3-4 points on the scale. In the text, the authors claim that there are 4-5 in the group. How can the authors explain this contradiction?
5. The authors in the text refer to Supplementary Figure S14, but I did not find this.
6. Abbreviations are not deciphered in Figure 3.
Minor:
- “Prospective, multicenter, ambulance-based, prospective cohort study...” (lines 22-23) repeat the word “prospective” twice.
- “A prospective, multicenter, ambulance-based, multicenter, prospective cohort study was undertaken...” (lines 70-71) repetition of the words “prospective” and “multicenter” twice.
References:
1. Pedersen CK, Stengaard C, Bøtker MT, Søndergaard HM, Dodt KK, Terkelsen CJ. Accelerated -Rule-Out of acute Myocardial Infarction using prehospital copeptin and in-hospital troponin: The AROMI study. Eur Heart J. 2023 Oct 12;44(38):3875-3888. doi: 10.1093/eurheartj/ehad447.
2. Aarts GWA, Camaro C, van Royen N. Ready for rapid rule-out of acute myocardial infarction. Eur Heart J. 2023 Oct 12;44(38):3889-3891. doi: 10.1093/eurheartj/ehad519.
3. Stopyra JP, Snavely AC, Scheidler JF, Smith LM, Nelson RD, Winslow JE, Pomper GJ, Ashburn NP, Hendley NW, Riley RF, Koehler LE, Miller CD, Mahler SA. Point-of-Care Troponin Testing during Ambulance Transport to Detect Acute Myocardial Infarction. Prehosp Emerg Care. 2020 Nov-Dec;24(6):751-759. doi: 10.1080/10903127.2020.1721740.
Comments on the Quality of English LanguageNo comments
Author Response
Dear Editor,
We would like to acknowledge the thorough revision of our work by the editor and reviewers. We also thank you for the opportunity to resubmit our work. We have responded to all the reviewers’ comments. Below, please find point-by-point responses coloured in red. All the comments that needed a change in the manuscript have been included in the main document and are also colored in red. Additionally, the whole text has been reviewed by the AJE editing service.
We believe that the manuscript has been substantially improved, and we hope that it can be considered for publication.
Kind regards,
Francisco Martín-Rodríguez on behalf of all the authors
Reviewer #2
I reviewed with interest the manuscript by Daniel Zalama-Sánchez et al, “Prehospital targeting of long-term mortality in acute chest pain by cardiac biomarkers.” In this article, the authors showed that a risk score based on prehospital cTnT, NT-proBNP, and D-dimer was able to predict long-term mortality in patients with acute chest pain.
This is probably an interesting result that expands the possibilities of prehospital diagnostics. However, during the review, I had questions and comments to which I would like to receive answers from the authors:
1. First of all, I have doubts about the basic concept of the work. In patients with acute chest pain, rapid diagnosis is required, and if acute coronary syndrome is detected, emergency treatment is required. Therefore, prehospital diagnostics should be aimed precisely at this, for example, this was implemented in the AROMI study (ref. 1-32, see below). Identifying prehospital predictors of 1-year mortality in patients with acute chest pain seems far-fetched. If there is no acute coronary pathology, then this can be safely done during an outpatient examination in primary care. I also disagree with the use of the term “long-term mortality” in relation to annual mortality.
Response: We would like to acknowledge the opportunity that the reviewer gave us to clarify this point. First, it should be noted that all the patients included presented with acute chest pain, and the procedure recommended by the reviewer (“rapid diagnosis is needed, and if acute coronary syndrome is detected, emergency treatment is required”) is the correct procedure. To address the 1-year mortality, we have included the following paragraph at the end of the discussion, previous to the limitations section. “Currently, it is not easy to determine 1-year outcomes since hidden pathologies or secondary consequences could emerge after an acute event. Therefore, extending the clinical decision-making process for early warning scores from short-term to 1-year outcomes could be particularly helpful for planning long-term care once the acute phase of the disease has ended or even for activating follow-up programs in the health system since the patient is potentially fragile.”
To cite previous works, as stated by the reviewer, we have included them following reference 27.
We have changed all instances of “long-term mortality”, which refers to 1-year mortality, to “1-year mortality”.
2. The authors submitted the text of the manuscript with highlighted changes after editing (apparently, according to previous reviews). This is unnecessary at this stage.
Response: This has been resolved.
3. The purpose of duplicating the block diagram (Fig. 1) is unclear to me.
Response: This was due to the previous revision. This has been solved.
4. In Figure 2a, visually the highest percentage of mortality was in the group with 3-4 points on the scale. In the text, the authors claim that there are 4-5 in the group. How can the authors explain this contradiction?
Response: We have corrected this mistake in the Results section. Since the reviewer is correct, there was a higher percentage of mortality at points 3-4. This is probably caused by the lower number of patients included in the 4-5 group.
5. The authors in the text refer to Supplementary Figure S14, but I did not find this.
Response: This was due to the previous revision. This has been solved.
6. Abbreviations are not deciphered in Figure 3.
Response: Abbreviations for both Figure 3 and 4 have been included.
Minor:
- “Prospective, multicenter, ambulance-based, prospective cohort study...” (lines 22-23) repeat the word “prospective” twice.
Response: Solved.
- “A prospective, multicenter, ambulance-based, multicenter, prospective cohort study was undertaken...” (lines 70-71) repetition of the words “prospective” and “multicenter” twice.
Response: Solved.
References:
1. Pedersen CK, Stengaard C, Bøtker MT, Søndergaard HM, Dodt KK, Terkelsen CJ. Accelerated -Rule-Out of acute Myocardial Infarction using prehospital copeptin and in-hospital troponin: The AROMI study. Eur Heart J. 2023 Oct 12;44(38):3875-3888. doi: 10.1093/eurheartj/ehad447.
2. Aarts GWA, Camaro C, van Royen N. Ready for rapid rule-out of acute myocardial infarction. Eur Heart J. 2023 Oct 12;44(38):3889-3891. doi: 10.1093/eurheartj/ehad519.
3. Stopyra JP, Snavely AC, Scheidler JF, Smith LM, Nelson RD, Winslow JE, Pomper GJ, Ashburn NP, Hendley NW, Riley RF, Koehler LE, Miller CD, Mahler SA. Point-of-Care Troponin Testing during Ambulance Transport to Detect Acute Myocardial Infarction. Prehosp Emerg Care. 2020 Nov-Dec;24(6):751-759. doi: 10.1080/10903127.2020.1721740.

Round 2
Reviewer 2 Report (New Reviewer)
Comments and Suggestions for Authors
The authors answered my questions and made corrections to the text. However, I have doubts about the scientific and practical value of the study.
Comments on the Quality of English LanguageNo comments
This manuscript is a resubmission of an earlier submission. The following is a list of the peer review reports and author responses from that submission.
Round 1
Reviewer 1 Report
Comments and Suggestions for Authors
The timely identification and management of patients at risk of acute chest pain (ACP), with or without myocardial infarction coding (MIC), can be an effective prehospital tool to aid patient triage during conveyance to the emergency department and streamline diagnosis and treatments, and patient care according to anticipated severity. In this manuscript, the authors propose the establishment of a scoring system based on 3 cardiac enzymes, troponin T (cTnT), N-terminal pro B-type natriuretic peptide (NT-proBNP), and D-dimer, to predict 1-year mortality outcomes. The authors split the total sample into 2 parts: two-thirds for derivation followed by one-third for validation. The combined score matrix presented an AUC of 0.802, suggesting that their model has good discriminatory power to identify high scoring patients at risk of long-term mortality. The work presented here is in line with the authors' research interests in prehospital point-of-care testing and has the potential to improve patient management, risk stratification and better medical resource allocation. I have the following comments to be addressed for publication considerations:
Major Comments:
1) Table 1 (Prehospital diagnosis) - p-value = 0.021 is awarded to ACP, what about MIC? or is this p-value generated for proportion of the two prehospital diagnosis (ACP and MIC) - in this case p-value should be assigned to prehospital diagnosis and not to ACP only. This data is also repeated in Table 2 unnecessarily.
2) The authors mentioned that total sample size were split into 2: for derivation and validation studies. However, it is unclear how these 2 study cohorts were different as the tables were pooled for total baseline characteristics. Were there differences in characteristics despite randomization? Suggest to portray this data in the supplementary materials for clarity.
3) Table 2 - I'm surprised that SpO2, % is stat sig. when median +/- IQR values are highly overlapping. Why may this be - sample size diff or proportions at different ends of the range? This is the same for ACCI. Please check the statistics performed are correct throughout. Also, Hospitalization days in MIC should be a typo: 5 (3-9) and not 5 (3.9)?
4) Table 3 - It is not clear how the designated concentrations and scoring weightage allocated were designed/generated. e.g. NT-proBNP only had the option of 0 or 2. Technically, the max score should be 6 and not 5 as presented in Figure 2 i.e. score 2 for each cTnT, NT-proBNP, D-dimer, or was there no such observation?
5) Figure 3G - It is surprising that RCO curve for patients (>74 years) is less predictive than that of (50-74 years). Also, the RCO doesn't seem predictive for younger patients (95% CI is variable). Similarly with the additional clinical variables of gender. Similarly, MIC patients have better predictability than ACP patients, supposedly due to greater severity. Seems like the authors presented but skimmed over these details without communication. There is a lack of explanation/discussion as to what these variables mean and what implications it may play in regards to the utilization of the scoring matrix.
6) What details are provided in the calibration plots (Figure 4 and 6) and how do they support the ROC curves? No emphasis has been made to these calibration curves beyond the statement "...results were supported by the calibration curves,...". They also distract from the main figures, without considerable focus on the results of these calibration curves. Suggest to move to supplementary figures.
7) Figure 5 - Has the authors explored if combination of all 3 biomarkers are actually better than combination of any 2? D-dimer alone does not seem to be very useful AUC is close to 0.5, suggesting it has limited predictive power on its own. In addition, AUC curves should be provided with p-values, as previously presented by the authors (PMID 34224955/ 35716940).
8) Results (Line 239) - There is no "Supplementary Figure 6", but a main Figure 6 is portrayed for calibration plots below Figure 5. Figure 6 image quality is poor and illegible. Please improve the quality.
9) Discussion (Line 282-4) - "Ratings ≥ 4 points correlate with a significantly high rate of advanced life support interventions, ICU admissions, and short- and long-term mortality. A rating between 0 and 1 point suggests a low risk of 1-year mortality". Is this true or proposed/suggested? The authors have not shown such data here, nor cited relevant data suggesting this. The ROCs generated in this paper are predictive for 1yr mortality only - what is the breakdown?
Minor Comments:
1) Title page affiliations - Drs del Pozo Vegas and Sanz-García are joint "last" not "first" authors
2) Abstract (line 23) and Study design (line 71) - repeated "multicenter". There is no mention how many and which hospitals in the province of Valladolid (Spain) was involved, and if ethics approval for all sites were sought. The authors claim "... was conducted after approval by the respective local ethics committees" (line 81), however only the IRB approval ID for Hospital Universitario Rio Hortega was provided (line 336).
3) Methods (line 140) and Figure 6 legend - new introduction of unconventional abbreviations for troponin T (cTnT), N-terminal pro B-type natriuretic peptide (NT-proBNP), and D-dimer. Please standardize.
4) Figure 1 Exclusion criteria - "Non interested in the study" should be rephrased to "unwilling to participate". The numbers do not add up to total excluded (n=1355) and (n=57) - please clarify.
5) Methods (line 149-151) - "Further details on the score development and validation methods can be found in the supplementary data" and "Further details on data collection, missing values, and sample size calculations can be found in the supplementary data". However, the document uploaded is not in English which precludes effective review.
6) In-text formatting - ensure space between table headings/footnotes and subsequent paragraphing and vice versa e.g. (Line 208-213).
7) Figure 3 - plot sequence should be consistent with flow of data presentation as has been demonstrated in tables earlier i.e. ACP should come before MIC.
Comments on the Quality of English LanguageGenerally fine.
Author Response
attachment

Reviewer 2 Report
Comments and Suggestions for Authors
Review:
The authors present an interesting study on a scoring system, in the prehospital setting, when acute chest pain is referred.
The topic is interesting, and the results are consistent with the analysis carried out. Diagnostics is appropriate for this article.
Some minor issue should be addressed before publication.
- A revision of the text is necessary to fix some typos (i.e. line 71 multicenter is repeated)
- A sample size estimation should be reported in the Methods (even if the authors discuss the limited size as a limit of the study)
- A discussion on the differences found on the AUC of mortality at 1-year for the ages considered (Fig. 3 e,f,g) should be welcome.

The quality of English language is sufficient for the present manuscript
Author Response
attachment
